# Avoiding, Not Managing, Drug Withdrawal Syndrome in the Setting of COVID-19 Acute Respiratory Distress Syndrome. Comment on Ego et al. How to Manage Withdrawal of Sedation and Analgesia in Mechanically Ventilated COVID-19 Patients? *J. Clin. Med.* 2021, *10*, 4917

**DOI:** 10.3390/jcm11123336

**Published:** 2022-06-10

**Authors:** Fabrice Petitjeans, Marc de Kock, Marco Ghignone, Luc Quintin

**Affiliations:** 1Critical Care, Hôpital d’Instruction des Armées Desgenettes, 69008 Lyon, France; fabricepetitjeans@yahoo.fr; 2Critical Care, Centre Hospitalier de Wallonie Picarde, 7500 Tournai, Belgium; marcdekock1888@gmail.com; 3Critical Care, JF Kennedy North Hospital, West Palm Beach, FL 33407, USA; torchio@aol.com

The management of sedation in the setting of COVID-19 (“COVID”) by Ego et al. [1] does not combine pathophysiology and pharmacology. Their premise rests on «*decreasing the work of breathing, applying lung protective ventilation and limiting asynchronies [to] minimize the risk of ventilator-induced lung injury (VILI)……COVID-19 patients require high [doses] of sedatives, analgesics and neuromuscular blocking agents (NMBA)……frequently for more than 7 days*» [1]. Ego manages the drug withdrawal syndrome but does not *avoid* it.

First, the requirements allowing for optimal ventilation in the setting of acute respiratory distress syndrome (ARDS), delineated earlier [2,3,4], are not addressed: *(Vt, f) = f(temperature, agitation, inflammation, lung water, pH, microcirculation, PaCO_2_, PaO_2_, positioning)*. Briefly, temperature is lowered to low normal (35–36 °C). Alpha-2 agonists suppress the tonic activity of the dorsal noradrenergic bundle [5], control agitation and avoid emergence delirium and withdrawal syndrome [6] («cooperative » sedation from endotracheal intubation onward, i.e., alpha-2 agonist as *first*-line sedative [7]: clonidine 2 μg kg^−1^ h^−1^ or dexmedetomidine 1.5 μg kg^−1^ h^−1^). To achieve −2 < RASS < 0 (stringent restlessness), alpha-2 agonists are supplemented with neuroleptics, if required (appendix in [8]), as in refractory delirium tremens [9]. Both drugs do *not* depress respiratory genesis [10]. Thus, conventional sedation is *not* needed following intubation. Adequate iterative circulatory optimization combined to the sympatholysis evoked by alpha-2 agonists normalizes the microcirculation, systemic pH, lactate concentration, CO_2_ gap and venous O_2_ saturation. Alpha-2 agonists present anti-inflammatory properties [11], either at the systemic or central nervous system or lung (tissue or receptors) level. In turn, normalized microcirculation eases diapedesis and improves the innate immune function: a return to normal functioning of the adrenergic receptors of immune cells possibly occurs (“upregulation”).

Second, alpha-2 agonists act via the sympathetic and the parasympathetic systems, beginning with intubation: cooperative sedation [12], with improved cognition [13,14], diuresis, lowered VO_2_ and inflammation, etc. As alpha-2 agonists evoke indifference to the environment and pain, opioids are counterproductive. Should the patient need analgesia, opioid free analgesia (appendix in [8]) does not depress respiratory genesis. Consequently, the duration of paralysis is reduced to a few hours in the setting of conventional ARDS (e.g., aspiration, etc.). Once the vicious circle of self-induced lung injury (SILI) is broken, spontaneous breathing resumes (e.g., pressure support delineated in [2,3,4]). PEEP is adjusted to a high level if diffuse ARDS is present. Upright position is set, meticulously.

*Early* COVID-ARDS presents with a high VA/Q ratio (lowered perfusion with near-normal ventilation, compliance, and lung mechanics). The inflammation of the lung capillaries and receptors and alveoli is addressed non-specifically. As COVID-19 does not weaken the ventilatory muscles, and as compliance is relatively high, brief paralysis *just* breaks the SILI and the high inspiratory drive. Spontaneous breathing avoids ventilator-induced lung injury. First-line, high-dose alpha-2 agonists combined to low normal temperature and normalized inflammation do not lead to «*high regimen and prolonged use of sedative, analgesics and neuromuscular agents*» [1]. In our hands [3], breaking up the SILI is achieved within 2 days with a low toll (mortality: 8.5% [3]), at variance with general anesthesia, paralysis and proning for weeks with critical care clogging and societal consequences. This [3,4] requires demonstration.

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
