# Peer review of "Avoiding, Not Managing, Drug Withdrawal Syndrome in the Setting of COVID-19 Acute Respiratory Distress Syndrome. Comment on Ego et al. How to Manage Withdrawal of Sedation and Analgesia in Mechanically Ventilated COVID-19 Patients? J. Clin. Med. 2021, 10, 4917"

_jcm, 2022, doi:10.3390/jcm11123336_

Round 1

Reviewer 1 Report

This paper focuses on a sedation and sedative-weaning strategy in covid-19 patients with ARDS requiring mechanical ventilation. While the strategy is interesting, and does contrast with the one by Ego as described in the first paragraph, without including even a case report, it seems as though the strategy, at this juncture, is theorized as being effective, and not the result of clinical trial or case experience.  Would help to report trial of the strategy and how it was effective in patients to be of clinical significance or to affect patient care.  The manuscript does have a fair number of grammatical errors as well and could benefit from some revision.